# High-resolution regional inversion reveals overestimation of anthropogenic 1 methane emissions in China 2 Shuzhuang Feng<sup>1,2</sup>, Fei Jiang<sup>1,2,5\*</sup>, Yongguang Zhang<sup>1,2</sup>, Huilin Chen<sup>3</sup>, Honglin Zhuang<sup>1</sup>, Shumin 3 Wang<sup>3,4</sup>, Shengxi Bai<sup>1</sup>, Hengmao Wang<sup>1,2</sup>, Weimin Ju<sup>1,2</sup> 4 5 <sup>1</sup> Jiangsu Provincial Key Laboratory of Geographic Information Science and Technology, 6 International Institute for Earth System Science, Nanjing University, Nanjing, 210023, China 7 <sup>2</sup> Jiangsu Center for Collaborative Innovation in Geographical Information Resource Development 8 and Application, Nanjing, 210023, China 9 <sup>3</sup> School of Atmospheric Sciences, Nanjing University, Nanjing, 210023, China 10 <sup>4</sup> Shanxi Province Institute of Meteorological Sciences, Taiyuan 030002, China 11 <sup>5</sup> Frontiers Science Center for Critical Earth Material Cycling, Nanjing University, Nanjing, 210023, 12 13 China 14 Corresponding author: Fei Jiang; jiangf@nju.edu.cn 15 16 17

Corresponding author: Fei Jiang; Tel.: +86-25-83597077; Fax: +86-25-83592288; E-mail address: jiangf@nju.edu.cn

## Abstract






















Methane (CH<sub>4</sub>), the second most important anthropogenic greenhouse gas, significantly impacts global warming. As the world's largest anthropogenic CH<sub>4</sub> emitter, China faces challenges in accurately estimating its emissions. Top-down methods often suffer from coarse resolution, limited data constraints, and result discrepancies. Here, we developed the Regional Methane Assimilation System (RegGCAS-CH<sub>4</sub>) based on the WRF-CMAQ model and the EnKF algorithm. By assimilating extensive TROPOMI column-averaged dry CH<sub>4</sub> mixing ratio (XCH<sub>4</sub>) retrievals, we conducted highresolution nested inversions to quantify daily CH<sub>4</sub> emissions across China, with a focus on Shanxi Province in 2022. Nationally, posterior CH<sub>4</sub> emissions were  $45.1 \pm 3.8$  TgCH<sub>4</sub>·yr<sup>-1</sup>, 36.5% lower than the EDGAR estimates, with the largest reductions in the coal and waste sectors. In North China, emissions decreased most significantly, mainly attributed to the coal and enteric fermentation sectors. Posterior emissions in coal-reliant Shanxi Province decreased by 46.3%. Sporadic emission increases were detected in major coal-producing cities but were missed by the coarse-resolution inversion. Monthly emissions exhibited a winter-low, summer-high pattern, with the rice cultivation and waste sectors showing higher seasonal increases than those in EDGAR. The inversion significantly improved XCH<sub>4</sub> and surface CH<sub>4</sub> concentration simulations, reducing emission uncertainty. Compared to other bottom-up/top-down estimates, our results were the lowest, primarily because the high-resolution inversion better captured local emission hotspots. Sensitivity tests underscored the importance of nested inversions in reducing the influence of boundary condition uncertainties on emission estimates. This study provides robust CH<sub>4</sub> emission estimates for China, crucial for understanding the CH<sub>4</sub> budget and informing climate mitigation strategies.






Keywords CH<sub>4</sub> emissions, Emission inversion, RegGCAS-CH<sub>4</sub>, Data assimilation


## 1. Introduction





























Methane (CH<sub>4</sub>) ranks as the second most significant anthropogenic greenhouse gas and plays a crucial role in global warming. Its global warming potential is 28–29.8 times that of carbon dioxide (CO<sub>2</sub>) over a 100-year time scale and contributes approximately 17% to the total radiative forcing of greenhouse gases since the industrial era (Forster et al., 2021; Saunois et al., 2025). With a relatively short steady state atmospheric budget lifetime of slightly over 9 years (Prather, 2007), CH<sub>4</sub> is a key target for rapid climate change mitigation efforts (Tu et al., 2024), as emphasized by the Global Methane Pledge, which aims to reduce global anthropogenic CH<sub>4</sub> emissions by 30% below 2020 levels by 2030 (GMP, 2023). China, as the world's largest CH<sub>4</sub> emitter, accounting for around 14%–22% of global anthropogenic CH<sub>4</sub> emissions, faces a complex challenge (Zhang et al., 2022; Janssens-Maenhout et al., 2019). China's CH<sub>4</sub> emissions originate from diverse sources, with coal mining and rice cultivation being particularly prominent (Nisbet, 2023; Lin et al., 2021). Comprehending the spatiotemporal distribution and trends of China's CH<sub>4</sub> emissions is of utmost importance for formulating effective climate policies and fulfilling international climate commitments. The bottom-up approaches, which rely on activity data and emission factors, have been widely used to estimate China's CH<sub>4</sub> emissions. However, accurately estimating these emissions remains a significant challenge. Existing inventories are often characterized by large uncertainties in both magnitude and sectoral attribution, with differences between various bottom-up estimates reaching up to 40–60% for China (Saunois et al., 2025). Among the various sources, estimates of China's coal mine CH<sub>4</sub> emissions can range from 14-28 Tg·yr<sup>-1</sup> (Sheng et al., 2019). This wide disparity leads to much uncertainty in the bottom-up estimates. There are multiple reasons for the uncertainties, especially the lack of comprehensive data on emission sources, especially for small-scale and sporadic emitters, and the use of complex and perhaps inappropriate emission factors in different sectors (Stavert et al., 2022). For instance, in coal mining, the emission factors vary significantly depending on the mining method (underground vs. surface mining), geological conditions, and the quality of coal. More generally, due to the use of a spatial proxy approach for the spatial allocation of the total emissions, existing global inventories exhibit significant spatial discrepancies when compared with high-resolution local measurements, resulting in misattribution of emissions (Qin et al., 2024).

In contrast, top-down inversion methods, that utilize satellite and surface observations in conjunction with atmospheric transport models, have the potential to provide more accurate and comprehensive estimates. Compared with ground-based inversions, satellite-based atmospheric inversions, such as those using column-averaged dry CH<sub>4</sub> mixing ratios (XCH<sub>4</sub>) data from the Greenhouse Gases Observing Satellite (GOSAT) or the TROPOspheric Monitoring Instrument (TROPOMI), can offer valuable insights into the spatial distribution of emissions (Lu et al., 2023). Studies utilizing satellite observations have quantified CH<sub>4</sub> emissions from various sources, including oil, gas, and coal mining sectors at the global, regional, and local scales (Nesser et al., 2024; Bai et al., 2024; Zhang et al., 2021). For instance, Maasakkers et al. (2019) investigated the contribution of different regions to the global CH<sub>4</sub> budget with GOSAT XCH<sub>4</sub> data, identifying areas with significant emissions and sinks. Pandey et al. (2019) used TROPOMI XCH<sub>4</sub> observations to reveal extreme CH<sub>4</sub> leakage from a natural gas well blowout, demonstrating the instrument's ability to detect both large-scale and short-term emission events. The Methane Emission Control Action Plan released by China explicitly states the exploration and implementation of research on atmospheric CH<sub>4</sub> emission inversion models, and strengthens the verification of emission inventories by inversion data (MECAP, 2023). A number of inversions relevant to China have been conducted with satellite observations. Using inverse analysis of 2019 TROPOMI XCH<sub>4</sub> data, Chen et al. (2022) quantified CH<sub>4</sub> emissions across China and attributed contributions to specific sectors. Zhang et al. (2022) estimated China's CH<sub>4</sub> emissions from 2010 to 2017 by combining satellite and surface observations, revealing complex linkages between emission trends and associated policy drivers. However, the trends of China's CH<sub>4</sub> emissions quantified by the top-down approach (Sheng et al., 2021; Miller et al., 2019) are contrary to those estimated by the bottom-up approach. This discrepancy is mainly due to the uncertain quantification of emissions from the coal mining sector (Sheng et al., 2019; Liu et al., 2021). Currently, global-scale CH<sub>4</sub> assimilation systems are widely applied, such as CarbonTracker-CH<sub>4</sub> in the United States (Bruhwiler et al., 2014), CAMS in Europe (Agustí-Panareda et al., 2023), NTFVAR in Japan (Wang et al., 2019), and GONGGA-CH<sub>4</sub> in China (Zhao et al., 2024). However, significant knowledge gaps remain in accurately estimating CH<sub>4</sub> emissions at the regional scale. There are relatively few existing regional CH<sub>4</sub> assimilation systems, such as the ICON-ART-CTDAS (Steiner et






























al., 2024) and CarbonTracker Europe-CH<sub>4</sub> (Tsuruta et al., 2017) in Europe. Additionally, several opensource frameworks offer inversion tools adaptable to different scales, such as LMDz-SACS-CIF in France (Thanwerdas et al., 2022) and the IMI in the United States (Varon et al., 2022). Nevertheless, most existing regional inversions still rely on global atmospheric transport models with relatively coarse resolutions and only provide annual or monthly average results. This lack of high spatiotemporal resolution prevents them from capturing local emission characteristics and short-term variations (Chen et al., 2022). Qu et al. (2021) highlighted significant challenges in separating rice and coal emissions over southeast China due to coarse grid resolution. High-resolution estimates are crucial for understanding the detailed distribution of emissions, especially in regions with heterogeneous source landscapes. For instance, in Shanxi Province, a major coal producing region, the traditional low-resolution models may fail to capture the emissions from numerous small-scale coal mines. Additionally, the inversion results of global and regional systems still show significant differences, mainly due to different observation data, inversion methods, transport models, and resolutions (Kou et al., 2025; Chen et al., 2022; Liang et al., 2023). Another challenge is that regional models need to consider the impact of boundary fields, especially for long-lived species. Current studies usually directly derive regional boundary fields from global simulation or analysis fields, which still contain large errors (Zhang et al., 2022; Kou et al., 2025). Consequently, China's contribution to the global CH<sub>4</sub> budget remains unclear. In this study, we aimed to address these limitations by developing a Regional Methane Assimilation System (RegGCAS-CH<sub>4</sub>) based on the Weather Research and Forecasting-Community Multiscale Air Quality (WRF-CMAQ) model and the Ensemble Kalman Filter (EnKF) algorithm (Evensen, 1994). This system enabled the assimilation of a large volume of TROPOMI XCH4 data to achieve nested inversions of daily CH<sub>4</sub> emissions with high spatial-temporal resolution. We conducted a comprehensive analysis of the spatial characteristics and monthly variations of China's CH<sub>4</sub> emissions in 2022. In particular, we focused on the nested inversion analysis of CH<sub>4</sub> emissions in Shanxi Province, China. The novelty of our study lies in the high spatial resolution emission inversions, which allows for






























capturing the fine-scale features of CH<sub>4</sub> emissions that are often missed by global inversion models.

Secondly, by assimilating a larger amount of TROPOMI XCH4 observations, which feature high

spatial resolution, wide spatial coverage, and high-frequency retrievals (~ 100 times more observations than GOSAT) (Qu et al., 2021), we can incorporate the latest and more representative information on atmospheric CH<sub>4</sub> concentrations to update the daily emissions. This high temporal resolution is essential for understanding the short-term fluctuations in emissions and their response to various factors. Notably, we have pre-updated the global boundary fields, which effectively mitigates the impact of boundary condition errors on regional emission inversions. This is a crucial step that has not been emphasized or implemented in most previous studies. By comparing our results with the currently widely used inventories, we aim to provide a more accurate and detailed understanding of China's CH<sub>4</sub> emissions, which is crucial for formulating effective climate change mitigation strategies.

## 2. Method and data





























### 2.1 Data assimilation system

The basic framework of the RegGCAS-CH<sub>4</sub> system is almost identical to that of RegGCAS (Zhang et al., 2024). Developed by Feng et al. (2020) based on the Regional multi-Air Pollutant Assimilation System (RAPAS, Feng et al., 2023), RegGCAS was initially used to infer the fossil fuel CO<sub>2</sub> emission. RegGCAS-CH<sub>4</sub> includes a regional chemical transport model (CTM) and an ensemble square root filter (EnSRF) assimilation module (Whitaker and Hamill, 2002), which are employed to simulate atmospheric compositions and infer anthropogenic emissions, respectively. In this study, the system was extended to high-resolution nested inversion for CH<sub>4</sub> emissions, which can optimize emissions from the outer (D01) to inner (D02) domain and reduce the influence of inaccurate boundary conditions on the inversion of the inner area (Feng et al., 2022). Moreover, the assimilation framework was updated to optimize CH<sub>4</sub> emissions by assimilating the TROPOMI CH<sub>4</sub> column retrievals. For the same domain, the RegGCAS-CH<sub>4</sub> performed a "two-step" inversion scheme in each data assimilation (DA) window. First, the prior emissions were optimized using the available atmospheric observations. Then, the optimized emissions were input back into the CTM to generate the initial fields for the next assimilation window. Simultaneously, the optimized emissions were transferred to the next window to serve as prior emissions (Figure S1). It is noted that the system optimizes the prior emissions for the D01 and D02 domains separately. Specifically, D01 only provides an optimized boundary field for D02, rather than the prior emission source for D02. Thus, the uncertainties in boundary conditions for D02 emission estimates were reduced. This "two-step" scheme facilitates error propagation and iterative emission optimization, which can ensure the mass conservation of the system and effectively enhance the stability and consistency of the emission updates (Feng et al., 2024).

## 2.1.1 Atmospheric transport model

The Weather Research and Forecast (WRF v4.0) model (Skamarock and Klemp, 2008) and the Community Multiscale Air Quality Modeling System (CMAQ v5.0.2) (Byun and Schere, 2006) were applied to simulate meteorological conditions and atmospheric chemistry, respectively. In this study, the CMAQ model employed two-nested simulations. The D01, which covered the whole mainland of China with a grid of 225 × 165 cells, and the D02, which covered the Shanxi Province and surrounding areas with a grid of 195 × 174 cells, had grid spacings of 27 and 9 km, respectively (Figure 1). There were 20 levels on the sigma–pressure coordinates extending from the surface to 100 Pa. To account for the rapid expansion of urbanization, we updated the underlying surface information for urban and built-up land using the MODIS Land Cover Type Product (MCD12C1) Version 6.1 of 2022. The detailed configuration of WRF-CMAQ is shown in Table S1.

**Figure 1** (a) Nested inversion domain and (b) number of TROPOMI XCH<sub>4</sub> retrievals during 2022. The red dashed frame depicts the CMAQ modeling domain; black squares represent the surface meteorological measurement sites; red triangles represent the six *in-situ* CH<sub>4</sub> measurement sites in Shanxi Province; purple asterisks and black triangles represent the flask and *in-situ* CH<sub>4</sub> measurement sites, respectively. The TROPOMI observations that fall within the same model grid are processed and counted as one super-observations. The subfigure in panel (b) shows the total number of super-observations for each month within the study area.

The meteorological initial and lateral boundary conditions were obtained from the Final (FNL) Operational Global Analysis data of the National Center for Environmental Prediction (NCEP) with a  $1^{\circ} \times 1^{\circ}$  resolution at 6-h intervals. The chemical lateral boundary conditions for the outer domain were derived from the CAMS global inversion-optimized CH<sub>4</sub> concentrations with a  $1^{\circ} \times 1^{\circ}$  resolution at 6-hour intervals (Bergamaschi et al., 2013), while those for the inner domain were obtained from the forward simulation of the outer domain with optimized CH<sub>4</sub> emissions. In the first DA window, the chemical initial conditions were also extracted from the CAMS, whereas in subsequent windows, they were derived through forward simulation using optimized emissions from the previous window. Given that the transport time of CH<sub>4</sub> within the study area is far shorter than its atmospheric lifetime, we deactivated all atmospheric chemical reaction processes to minimize computational costs (Chen et al., 2022; Kou et al., 2025). Instead, we integrated a set of CH<sub>4</sub> tracer variables into CMAQ for ensemble simulations, enabling all CH<sub>4</sub> concentration sets to be obtained from a single simulation.

Eliminating biases in boundary conditions is critical, as such biases can propagate through the entire system. We found that the boundary conditions extracted from the CAMS global fields still had considerable biases over East Asia (see Section 4). Thus, we calculated a grid-specific scaling factor for CAMS fields (50°E–160°E, 0°–70°N) against TROPOMI XCH<sub>4</sub> retrievals, and then applied these factors to correct the CAMS boundary conditions (Figure S2). Based on each pair of CAMS and TROPOMI data, we first calculated the column concentration of the CAMS reanalysis field. Then, the column concentrations of CAMS and TROPOMI were respectively smoothed in the longitudinal and latitudinal directions with a radius of 4°, and in terms of time using a 4-day window. Subsequently, the latitudinal average values were calculated, along with the average column concentrations biases between CAMS and TROPOMI. After that, linear interpolation was applied to fill in the missing values of the biases in the longitudinal direction. For the biases of the grids where there were missing values at the same latitude, it was assumed that they were consistent with the average biases of the non-missing values at that latitude. Finally, grid-by-grid bias was calculated and correction was carried out for the original CAMS CH<sub>4</sub> concentrations.

#### 2.1.2 EnKF assimilation algorithm

The EnKF is based on the Monte Carlo approach, which uses a stochastic ensemble of model states to approximate the probability distribution of the true state. It has been widely employed for updating the

model state by incorporating observational data to minimize the difference between the modelsimulated and observed values (Evensen, 1994). The ensemble square root filter (EnSRF) approach, introduced by Whitaker and Hamill (2002), was used to constrain the CH<sub>4</sub> emissions in this study. The EnSRF process commences with the initialization of an ensemble of model states, which are generated by applying a Gaussian perturbation with an average value of zero and the standard deviation of the uncertainty to a state vector  $X^b$ . The ensemble-estimated background error covariance matrix  $P^b$  is then calculated as:

$$P^{b} = \frac{1}{N-1} \sum_{i=1}^{N} (X_{i}^{b} - \overline{X}^{b}) (X_{i}^{b} - \overline{X}^{b})^{T}$$
 (1)














where N is the ensemble size;  $X_i^b$  represents the ith sampling;  $\overline{X}^b$  represents the mean of the ensemble samples.  $P^b$  plays a pivotal role in determining how the model state will be adjusted based on new observations.

During the forecast step, each ensemble member is advanced in time using the WRF-CMAQ model. As the model runs, uncertainties in emissions can lead to errors in CH<sub>4</sub> concentrations, and thus the response relationships of the concentration ensembles to the emission ensembles are obtained. In the analysis step, observational data y are incorporated to update the analyzed state. The ensemble mean of the analyzed state  $\overline{X}^a$  is regarded as the best estimate of emissions, which is obtained through the following equations:

$$\overline{X^a} = \overline{X^b} + K(y - H\overline{X^b}) \tag{2}$$

$$K = P^b H^T (HP^b H^T + R)^{-1}$$
(3)

where R is an observation error covariance matrix, which is specified as a diagonal matrix with the assumption that observation errors from different pixels are mutually independent (Feng et al., 2020). K is the Kalman gain matrix, estimated from the ensemble simulations and determining the relative contributions of observation and background to analysis. The state vector was defined as  $X^b = (Ea^T, Ep^T)^T$ , where Ea and Ep represent the vectors of CH<sub>4</sub> emissions for the area and power plant sources, respectively. Area sources included the daily total emissions from the enteric fermentation & manure, landfills & waste, rice cultivation, coal mining, oil & gas, industry, transport sources, etc.

Given that power plants are typically elevated point sources, this spatial distinction allows for effective separation from ground-based area sources. Therefore, even though power plant sources account for a small proportion (0.6%) of total emissions, we treated them as separate state vectors for optimization. The updated emissions are then used as the new initial states for the next forecast step, creating a cycle of assimilation that gradually refines the estimate of CH<sub>4</sub> emissions.

The observation operator H maps the model state to the observation space. In the context of CH<sub>4</sub>, H is configured to first horizontally geo-locate simulated CH<sub>4</sub> concentrations to match the TROPOMI XCH<sub>4</sub> retrievals. Subsequently, it remaps the sub-column concentrations from the 20-layer CMAQ vertical grid to the 12-layer TROPOMI vertical grid by totally or partially allocating CMAQ layers to TROPOMI layers based on pressure edges (Varon et al., 2022). Finally, the column average dry-air mixing ratio  $XCH4_s$  can be obtained by applying the TROPOMI column averaging kernel of each layer  $a_i$  to sub-columns:

$$XCH4_{s} = VCH4_{s}/VAIR_{dry,i} \tag{4}$$

where n is the number of retrieval layers;  $\Delta VCH4_{p,i}$  and  $VCH4_p$  represent the prior CH<sub>4</sub> column

$$VCH4_s = VCH4_p + \sum_{i=1}^{n} a_i \left( XCH4_{s,i} \Delta VAIR_{dry,i} - \Delta VCH4_{p,i} \right)$$
 (5)

in retrieval layer i and total CH<sub>4</sub> column;  $XCH4_{s,i}$  is the simulated dry air mixing ratio of CH<sub>4</sub> in retrieval layer i;  $\Delta VAIR_{dry,i}$  and  $VAIR_{dry,i}$  represent the dry air column in retrieval layer i and total dry air column, respectively, provided along with the TROPOMI product. In this study, the DA window was set to 1 d, meaning that daily TROPOMI XCH4 retrievals were utilized as emission constraints. The ensemble size was set to 50 to fully characterize the system uncertainties and inversion accuracy. To address the problem of spurious long-distance correlations in the EnKF (Houtekamer and Mitchell, 2001), we applied covariance localization using the Gaspari and Cohn function, which is a piecewise continuous fifth-order polynomial approximation of a normal distribution (Miyazaki et al., 2017; Gaspari and Cohn, 1999). Taking into account the transmission distance of CH<sub>4</sub> within one assimilation window, the localization scale was set to 300 km. By setting the covariance to be smaller the farther away from the observations, we could reduce the spurious influence of remote observations on the local analysis.

#### 2.2 Prior emissions and uncertainties

The prior anthropogenic CH<sub>4</sub> emissions were taken from the Emission Database for Global Atmospheric Research version 8.0 (EDGAR v8), which offers detailed monthly gridded CH<sub>4</sub> emissions at  $0.1^{\circ} \times 0.1^{\circ}$  from various anthropogenic sources (Crippa et al., 2024). The daily emissions, obtained by uniformly allocating the aggregated monthly emission inventory, were directly utilized as the initial estimate in the RegGCAS-CH<sub>4</sub>. For natural CH<sub>4</sub> emissions, we utilized the ensemble average of 18 emissions from the WetCHARTs v1.3.1 inventory ( $0.5^{\circ} \times 0.5^{\circ}$ , monthly) in 2019 for wetland emissions (Bloom et al., 2021), the Global Fire Emissions Database (GFED v4.1s,  $0.25^{\circ} \times 0.25^{\circ}$ , three-hourly) in 2022 for biomass burning emissions (van Wees et al., 2022), and CAMS global emissions inventory (CAMS-GLOB-TERM v1.1,  $0.5^{\circ} \times 0.5^{\circ}$ , monthly) in 2000 for termite emissions (Jamali et al., 2011). Additionally, CH<sub>4</sub> sinks resulting from soil absorption were derived from datasets simulated by Soil Methanotrophy Model (MeMo v1.0,  $1^{\circ} \times 1^{\circ}$ , yearly) in 2020 (Murguia-Flores et al., 2018). Given the model error compensation and the relatively comparable emission uncertainties from one day to the next, we applied an identical uncertainty of 40% to each emission grid at every DA window. Since the RegGCAS-CH<sub>4</sub> adopts a "two-step" inversion strategy and the daily posterior emissions are

#### 2.3 Assimilation data and errors

TROPOMI, onboard the Sentinel-5 Precursor satellite, was launched in 2017. Operating in a sunsynchronous orbit with an equator local overpass time of 13:30 h, TROPOMI offers daily global continuous monitoring of XCH<sub>4</sub>. The RemoTeC full-physics algorithm is used to retrieve XCH<sub>4</sub> from TROPOMI measurements of sunlight backscattered by Earth's surface and atmosphere in the near-infrared (NIR) and shortwave-infrared (SWIR) spectral bands (Lorente et al., 2022). After 2019, the spatial resolution of TROPOMI was adjusted to a remarkable 5.5×3.5 km<sup>2</sup> at nadir, enabling the identification of even relatively small-scale CH<sub>4</sub> emission sources. We used the TROPOMI XCH<sub>4</sub> level 2 data product to estimate CH<sub>4</sub> emissions.

iteratively optimized, the emission analysis is generally no longer sensitive to the prior uncertainties

of the original emission inventory after several assimilation windows (Zhang et al., 2024).

All TROPOMI individual pixels with a quality assurance value (qa\_value) smaller than 0.5 were discarded, which corresponds to high-cloud conditions or the presence of snow or ice. However, we still found many unrealistic low values, especially in summer. These negative biases can inevitably

lead to the underestimation of inverted emissions. To further minimize the impact of outliers, we selected another XCH<sub>4</sub> product generated based on the WFMD algorithm for cross-validation (Schneising et al., 2023). Only those pixels that were concurrently available in the TROPOMI/WFMD product and met the quality flag requirements were assimilated. Subsequently, the final qualitycontrolled TROPOMI data demonstrated good consistency with observations from two TCCON stations, namely the Hefei and Xianghe stations (Figure S3). Figure 1b illustrates the observational amount of TROPOMI XCH<sub>4</sub> in 2022 at each grid. Although the distribution of filtered data exhibits spatiotemporal nonuniformity, most grid cells in the central-northern regions with intense anthropogenic emissions have observational coverage for more than 100 days in 2022. Additionally, the monthly variation in the data amount shows fluctuations, with the peak occurring at the beginning and end of the year and the troughs around the middle. For regions with limited observation coverage (e.g., southern China), posterior emission estimates may rely heavily on prior information (see Discussion). On one hand, the system optimizes emissions in grids surrounding observations through the source-receptor relationship of atmospheric transport, allowing it to impose extensive constraints on emissions (Figure S4); on the other hand, it adopts an iterative approach where emissions optimized in the current window serve as prior emissions for the next window, facilitating rolling assimilation and thereby sustaining the influence of observational information on emission estimates. However, intermittent observations may cause posterior emissions to underestimate short-term emission fluctuations. At the monthly scale, grids without continuous observational constraints throughout the month directly use EDGAR data. Such grids account for 7.9% of all grids and contribute 0.3% to total posterior emissions. Although this may lead to insufficient observational constraints on posterior emissions, particularly in southern regions during summer, it effectively avoids seasonal distortions in posterior estimates caused by variations in emissions. At the annual scale, 4.8% of grids remain unadjusted. These unadjusted emissions are mainly distributed in uninhabited areas of Southwest China, resulting in a negligible overall impact on annual CH<sub>4</sub> emission estimates.






























According to the latest quarterly validation report, the  $1\sigma$  spread of the relative difference between the TROPOMI and the TCCON (Wunch et al., 2011) is of the order of 0.7% for bias corrected product, which is recommended to be considered as an upper boundary of the random uncertainty of the satellite data (Lambert et al., 2025).

#### 2.4 Evaluation Data










To evaluate the performance of the WRF simulations, we utilized the surface meteorological 322 measurements of 400 stations with 3-hour intervals, including temperature at 2 m (T2), relative 323 humidity at 2 m (RH2), wind direction at 10 m (WD10), and wind speed at 10 m (WS10). These 324 measurements were obtained from the National Climate Data Center (NCDC) integrated surface 325 database (http://www.ncdc.noaa.gov/oa/ncdc.html, last access: 25 August 2024). To evaluate the 326 327 posterior CH<sub>4</sub> emissions, two parallel forward modeling experiments were conducted: the control experiment with prior emissions (CEP) and the validation experiment with posterior emissions (VEP). 328 329 Both experiments utilized identical meteorological fields, as well as initial and boundary conditions. 330 We evaluated: (1) the simulated XCH<sub>4</sub> against TROPOMI XCH<sub>4</sub>; (2) the simulated surface CH<sub>4</sub> concentrations against independent ground CH<sub>4</sub> observations from eleven in-situ and five flask 331 monitoring sites (Table S2). The *in-situ* measurements included five global sites outside China (AMY, 332 333 GSN, RYO, ULD, and YON) with hourly resolution from NOAA's GLOBALVIEWplus CH<sub>4</sub> ObsPack v6.0 (Schuldt et al., 2023), as well as six regional sites in Shanxi Province (TY, DT, LF, SZ, JC, and 334 WTS) equipped with Picarro G2301 analyzers for high-precision, 5-second CH<sub>4</sub> measurements (data 335 were processed as daily averages to reduce random errors). The Picarro instrument, based on 336 wavelength-scanned cavity ring-down spectroscopy technology, is designated by the World 337 338 Meteorological Organization as the international reference instrument for CH<sub>4</sub> observations in 339 international comparisons. Additionally, observations from five flask sampling sites (AMY, DSI, LLN, TAP, and WLG) were also obtained from ObsPack dataset, providing weekly measurements. 340

## 3. Results and discussion

## 3.1 Posterior CH<sub>4</sub> emissions

Figure 2 shows the spatial distributions of the estimated annual CH<sub>4</sub> emissions and the differences from the prior emissions (i.e., EDGAR) over China in 2022. On a national scale, the high CH<sub>4</sub> emissions were primarily concentrated in the Yangtze River Delta (YRD), Pearl River Delta (PRD), North China Plain (NCP), and Shanxi Province. For the YRD and PRD characterized by high-density populations, a large number of industrial plants, as well as extensive agricultural activities such as rice cultivation, contributed significantly to CH<sub>4</sub> emissions. The NCP, with its heavy reliance on coal-based

energy and large-scale livestock farming, was also expected to be a major emission source. Lower emissions were mainly distributed across Northwest, and Southwest China.




















Compared with the EDGAR, the posterior emissions were generally smaller over most areas of mainland China, with total anthropogenic emissions decreasing to 45.1 TgCH<sub>4</sub>·yr<sup>-1</sup>, 36.5% lower than the EDGAR (71.0 TgCH<sub>4</sub>·yr<sup>-1</sup>). Additionally, previous studies have consistently shown that the EDGAR inventory overestimates China's CH<sub>4</sub> emissions. For example, through a Bayesian inversion of CH<sub>4</sub> and stable isotope ( $\delta^{13}$ C-CH<sub>4</sub>) measurements for East Asia, Thompson et al. (2015) found that posterior values decreased significantly across eastern and southern China, especially in the NCP. Overall, EDGAR overestimated China's emissions by 29%. Similarly, Zhang et al. (2021) conducted a global inverse analysis using GOSAT observations and revealed that EDGAR significantly overestimates anthropogenic emissions in eastern China. The posterior estimate of Chinese anthropogenic emissions was 30 % lower than that of EDGAR. Turner et al. (2015) also demonstrated that after assimilating GOSAT observations, China's posterior CH<sub>4</sub> emissions were revised downward by 50% relative to EDGAR. Using a regional model to assimilate TROPOMI observations, Kou et al. (2025) detected decreases of varying magnitudes across nearly the entirety of China, with the exception of Northwest China. Similar results have been reported by Alexe et al. (2015), Pandey et al. (2016), and Maasakkers et al. (2019). Our study, using a higher-resolution regional inversion method, provided more detailed emission information. Overall, our inversion results were comparable to the ensemble mean of GCP ground-based inversions (Wang et al., 2025).

**Figure 2** Spatial distribution of the annual total prior emissions (top, EDGAR v8, MgCH<sub>4</sub>·km<sup>-2</sup>·yr<sup>-1</sup>), posterior emissions (middle), and differences (bottom, posterior minus prior) over (a-c) Mainland China and (d-f) Shanxi Province.

Assimilating total CH<sub>4</sub> observations alone cannot disentangle emissions from different source sectors overlapping in individual grid cells (Saunois et al., 2025). Consequently, we partitioned the inversion results into respective emission sectors based on the monthly prior proportions at the model grid points

(Kou et al., 2025; Zhang et al., 2022), though this approach does introduce a certain degree of uncertainty in sectoral attribution. The sectoral patterns offer insights into the underlying factors influencing China's emission changes. We concentrated on interpreting the emissions from the coal, gas, rice cultivation, waste, livestock, building, and manure management sectors, which are the most significant sectors in China (Table S3). Table 1 shows the comparison of posterior and prior anthropogenic CH<sub>4</sub> emissions of the main emission sectors in China. Consistent with the previous studies, we also found that North China is the region with the most significant reduction nationwide. Almost the entire region has experienced a decrease, with a reduction of 56.9%, indicating that there are indeed substantial systematic biases in the EDGAR inventory. Among them, the coal sector contributed the most to this discrepancy, followed by enteric fermentation, with decreases of 56.2% and 64.0%, respectively (Table 1). For Northeast China, the increase in emissions mainly occurred in Heilongjiang Province, especially in the western part, which is the base of Daging Oilfield, China's largest oilfield. The oil and gas sectors in the entire Northeast China increased by 27.1% and 23.1%, respectively. However, emissions from other sectors decreased, resulting in an overall 11.1% reduction in emissions. Rice paddy CH<sub>4</sub> emissions serve as the dominant emission source in East China. However, the high spatial heterogeneity and the insufficiency of data on rice cultivation introduce large uncertainties to inventories. Additionally, different fertilization management practices in various regions, such as the use of nitrogen fertilizer versus organic fertilizer (Zhang et al., 2022), and mid-season drainage management (Lin et al., 2021), bring about considerable uncertainties in the emission factors related to rice cultivation practices. Emissions in Zhejiang, Fujian, and Jiangxi Provinces increased, mainly attributed to emissions from rice paddies. In contrast, emissions in other provinces decreased, dominated by the coal mining, leading to an overall 26.4% reduction in emissions in East China. Recent studies have highlighted significant errors in the spatial distribution of rice CH<sub>4</sub> emissions in EDGAR v8.0, which relies on outdated rice paddy maps and incorrectly overspreads rice emissions across non-rice agricultural grids (Chen et al., 2025). These limitations not only cause EDGAR to






























overestimate rice emissions but also lead to overestimation in subsequent posterior emissions.

Specifically, this overestimation may inflate the contribution of rice to total CH<sub>4</sub> emissions in posterior

attribution, while simultaneously underestimating the contribution of other CH<sub>4</sub> sources (e.g., coal,

wetlands) that coexist in these misclassified grids. Conversely, EDGAR fails to capture recent expansions of rice cultivation in Northeast China, particularly the rapid growth of rice paddies in the Sanjiang Plain (Liang et al., 2024). This omission may result in a systematic underestimation of rice emission hotspots in this region.

In Central and South China, the dominant sources of emissions remain rice paddies. However, the decreases were mainly attributed to the coal mining and wastewater treatment sectors, which reduced emissions by 51.3% and 12.4%, respectively. Overall, the total emissions in these two regions decreased by 21.8% and 7.8%, respectively. In Northwest China, the reduction in emissions was mainly distributed in the hotspot areas of coal mines, which dominated the overall 55.7% decrease in emissions. In Southwest China, the increase in emissions occurred predominantly in the southern part of the Sichuan Basin, a major coal-producing region, while emissions in other regions decreased. Sheng et al. (2019) found that the EDGAR inventory lacks a significant amount of statistics on coal mines. Moreover, China has issued an energy policy of "phasing out small coal mines" (approximately 50% of which are located in the southwest with high CH<sub>4</sub> content) to shift production towards lower-emission areas and consolidate into large coal mines (Zhang et al., 2022; Sheng et al., 2019). Additionally, the wastewater treatment sector achieved significant emission reductions, driving a 1.7% overall decrease in Southwest China.

**Table 1** Comparison of posterior and prior anthropogenic CH<sub>4</sub> emissions (GgCH<sub>4</sub>·yr<sup>-1</sup>) of the main emission sectors in the 7 major regions of China.

|           |           | Coal    | Gas    | Rice   | Waste  | Livestock | Building | Manure |
|-----------|-----------|---------|--------|--------|--------|-----------|----------|--------|
| North     | Prior     | 13841.1 | 98.4   | 169.1  | 2462.1 | 3882.9    | 385.5    | 859.5  |
|           | Posterior | 6068.4  | 46.2   | 116.4  | 1238.1 | 1396.4    | 206.7    | 279.6  |
| Northeast | Prior     | 771.9   | 82.6   | 879.6  | 1427.9 | 251.9     | 268.4    | 36.8   |
|           | Posterior | 665.0   | 101.8  | 821.2  | 1259.6 | 237.7     | 243.3    | 34.1   |
| East      | Prior     | 2257.3  | 54.4   | 5100.3 | 6681.4 | 907.9     | 732.9    | 200.5  |
|           | Posterior | 1114.5  | 45.1   | 4592.4 | 4750.9 | 512.7     | 576.2    | 114.4  |
| Central   | Prior     | 846.5   | 32.4   | 3139.5 | 1882.9 | 434.2     | 399.4    | 91.2   |
|           | Posterior | 412.2   | 25.7   | 2862.6 | 1324.4 | 343.3     | 314.1    | 75.2   |
| South     | Prior     | 15.0    | 20.5   | 2186.4 | 1622.5 | 280.2     | 254.6    | 44.0   |
|           | Posterior | 13.8    | 18.9   | 2080.6 | 1421.3 | 269.9     | 235.0    | 42.2   |
| Northwest | Prior     | 8188.3  | 1083.4 | 143.3  | 970.5  | 746.2     | 226.7    | 76.4   |
|           | Posterior | 2899.1  | 645.8  | 120.7  | 593.6  | 534.3     | 164.4    | 39.6   |
| Southwest | Prior     | 531.3   | 76.5   | 2046.4 | 1124.0 | 667.0     | 356.3    | 107.1  |
|           | Posterior | 505.6   | 77.2   | 2063.0 | 1060.7 | 663.7     | 352.1    | 107.2  |

<sup>\*</sup> Waste includes wastewater treatment, solid waste landfills, and solid waste incineration; Building represents emissions from small-scale non-industrial stationary combustion; Manure refers to emissions from the manure management sector.

China accounts for as high as 69% of the global mitigation potential in coal mining in 2030 (EPA, 2019). Shanxi Province, where 94% of the emissions come from the coal mining sector, accounts for nearly one-third of the country's total CH<sub>4</sub> emissions. Therefore, we conducted a focused analysis on the emissions in Shanxi Province with high-resolution (9 km) inversion, which can better diagnose the spatial characteristics of emissions caused by coal mining (Figure 2d-f). The optimized emissions showed a decrease in the vast majority of areas in Shanxi Province. Compared with the EDGAR inventory (9.0 TgCH<sub>4</sub>·yr<sup>-1</sup>), the posterior emissions decreased by 46.3% to 4.8 TgCH<sub>4</sub>·yr<sup>-1</sup>. The overestimation of coal mining emissions may be due to the fact that the standard IPCC emission factors used by EDGAR are too high for Chinese coal mines (Lin et al., 2021). Additionally, the average emission factors in the north are lower than those in other regions (Gao et al., 2021). The use of a

uniform regional emission factor by EDGAR further exacerbates the overestimation of emissions (Shi et al., 2025). Moreover, in the past decade, the extraction and utilization of coal mine CH<sub>4</sub> in China have been largely improved (Lu et al., 2021), but the recovery of coal mine CH<sub>4</sub> is not adequately considered in the EDGAR inventory. Finally, the CH<sub>4</sub> emission intensity of surface mining is ten times lower than that of underground mining. However, surface mining is overlooked in the EDGAR inventory (Gao et al., 2020). In Yangquan, Jincheng, Changzhi, and Jinzhong cities (Figure 2), some increased emission hotspots caused by the coal mining activities could be found. Coincidentally, these cities are the main coal-producing areas in Shanxi Province. Overall, the emissions in these cities have still decreased by 3.3%, 4.2%, 29.5%, and 32.3%, respectively (Figure S5). Tu et al. (2024) utilized TROPOMI observations and implemented a wind-assigned anomaly method to quantify the CH<sub>4</sub> emissions from coal mines in Yangquan, Changzhi, and Jincheng. Compared to EDGAR, emissions decreased by 56.5%, 40.5%, and 65.0%, respectively—a larger reduction than our findings.

## 3.2 Monthly variations

Figure 3 illustrates the comparison of the monthly variations in prior and posterior emissions both in China and Shanxi Province. Nationally, influenced by CH<sub>4</sub> emissions from paddy fields, both the prior and posterior emissions exhibited the characteristic of being low in winter and high in summer. However, the posterior emissions for each month were lower than the prior emissions. The monthly posterior emissions ranged from 2.3 to 7.5 Tg·month<sup>-1</sup>, with a modification rate ranging from -30.7% to 5.6% compared to the prior emissions (4.8–7.6 Tg·month<sup>-1</sup>). The month with the smallest difference occurred in August, mainly because the underestimation of CH4 emissions from paddy fields compensated for the overestimation of emissions from coal mining (Figure 3c). Additionally, due to the weather conditions such as clouds and rain in summer, the amount of TROPOMI data was significantly smaller than that in other seasons (Figure 1b), which might lead to insufficient constraints on emissions. Regarding Shanxi Province, even with an agnostic flat monthly prior (Figure 3b), our estimates generated a monthly variation, and emissions in all months were lower than the prior values. We also detected a decrease and subsequent increase in emissions that correspond to the Spring Festival in February. A noticeable recovery of production capacity in the following two months was also evident. Overall, although relatively large uncertainties were introduced by the amount of observations, the seasonal variation of monthly posterior emissions could be roughly captured.

It could be found that the reduction in emissions each month in China was mainly dominated by the coal sector, with an overall annual reduction of 55.9%, indicating a high level of uncertainty in the prior emissions. Moreover, the gas, livestock, and manure management sectors also showed varying degrees of reduction in different months, with overall decreases of 34.0%, 45.1%, and 51.5%, respectively. However, for the rice cultivation sector, the posterior emissions were lower than the prior emissions in the first half of the year, while the situation was reversed in the second half of the year. Summer and autumn are the heading and flowering stages for both single-cropping late rice and double-cropping late rice across the eastern, central, and southern China. During these periods, the emissions from paddy fields tended to be higher than in other seasons. However, EDGAR v8.0 adopts a uniform seasonal profile for rice CH<sub>4</sub> emissions across China, assigning a single emission peak in June to all rice-growing regions. This simplification contradicts the findings of Chen et al. (2025), who reported that rice CH<sub>4</sub> emissions in China generally peak in July-August, with the length of the emission season varying significantly due to the diversity of regional rice cropping systems. Notably, our posterior emission results align well with the seasonal pattern, with the highest monthly rice emissions occurring in August, followed by July (Table S3). This consistency confirms that the TROPOMI satellite observations have effectively corrected the unrealistic uniform seasonal bias inherent in EDGAR. Overall, the posterior CH<sub>4</sub> emissions from paddy fields have slightly decreased by 4.7%. A similar situation was observed in the waste sector. Previous studies have shown that significant CH<sub>4</sub>






























A similar situation was observed in the waste sector. Previous studies have shown that significant CH<sub>4</sub> production in wastewater is more likely to occur because methanogens become more active as the temperature rises (Hu et al., 2023). Therefore, a significant increase in the posterior emissions compared to the prior emissions could be observed in August. For the building sector, due to winter heating or hot water supply, the emissions in winter were approximately three times higher than those in summer. After optimization, the emissions in winter and spring significantly decreased, while in some months of summer and autumn, the decrease was less pronounced, and there was even a slight increase. The inter-monthly difference in emissions decreased, but the monthly variation remained significant. Overall, the posterior emissions have decreased by 25.4%. The changes in emissions in Shanxi Province were mainly dominated by the coal mining sector. Compared to EDGAR, the monthly reduction rate ranged from 12.6% in August to 67.3% in February. For other sectors, there was little













**Figure 3** Comparison of the monthly variations and the sectoral differences (GgCH<sub>4</sub>·mon<sup>-1</sup>) in prior and posterior CH<sub>4</sub> emissions over (a, b) China and (c, d) Shanxi Province.

## 3.3 Evaluation of posterior emission estimates

CH<sub>4</sub> emission estimates are highly sensitive to biases in meteorological simulations, as meteorological processes significantly influence atmospheric transport, which in turn shapes the source-receptor relationships and determines the flow-dependent background error covariance. Overall, the model demonstrated satisfactory performance in reproducing domain-wide meteorological conditions across China (Figure S6), with minimal biases of -0.4°C for T2 and -4.9% for RH2, alongside high correlation coefficients (CORR) of 1.0 and 0.96, respectively. The model's performance over Shanxi Province was slightly less optimal, likely due to the region's complex terrain. The biases for T2 and RH2 were 1.5°C and -12.5%, respectively, while the CORR remained high at 0.99 and 0.90. Additionally, the

WRF model effectively captured the temporal variations in wind direction both across China and Shanxi Province. However, WS10 was generally slightly overestimated, with biases of 0.5 m/s and 0.4 m/s and CORR of 0.88 and 0.81, respectively. Such overestimation of wind speed in WRF simulations has also been widely reported in other studies (Hu et al., 2016). An overestimated wind speed causes the model to simulate faster and more extensive diffusion of CH<sub>4</sub> concentrations than occurs in reality. To compensate for the simulated reduction in CH<sub>4</sub> concentrations due to this excessive diffusion, the inversion system potentially increases the estimated emissions.




























Figure 4 shows the spatial distribution of XCH<sub>4</sub> in the posterior simulation, as well as the validation of XCH<sub>4</sub> simulated by prior and posterior emissions with TROPOMI observations. It could be observed that relatively large XCH<sub>4</sub> were present over eastern China, which was driven by coal mining in the northern part and rice paddy fields in the southern part (Zhang et al., 2023). The simulation using prior emissions significantly overestimated the XCH<sub>4</sub> concentration in China, especially in the NCP. The maximum overestimation exceeded 100 ppb, which was consistent with the overestimated emissions in these regions (Figure 2). After the inversion optimization, the simulated XCH<sub>4</sub> showed better spatial distribution consistency with the TROPOMI observations. The vast majority of the biases were within 20 ppb, and the national average biases decreased from 14.7 ppb to 7.9 ppb, representing a 46.0% reduction. In Shanxi Province, high emissions were mainly concentrated in the coal-rich southeastern region, with an annual average maximum exceeding 1970 ppb, which was difficult to reproduce the characteristics of local high concentrations at a coarse resolution. The prior simulation showed obvious overestimation across the entire region, particularly in northern Shanxi. Due to the high spatial resolution adopted, the model could take into account more practical factors, and thus capture the spatiotemporal variations of emission sources and transport processes more precisely. As a result, the agreement between the posterior simulations and the TROPOMI observations was remarkably enhanced. The average deviation decreased from 37.9 ppb to 11.9 ppb, with a reduction of 68.7%. These changes not only demonstrate the effectiveness of reducing uncertainties in optimizing emission sources but also imply that there are indeed significant uncertainties in the prior emission inventory.

**Figure 4** Comparison of simulated XCH<sub>4</sub> (ppb) from prior and posterior emissions with TROPOMI observations over (a-c) China and (d-f) Shanxi Province for the 2022 annual average. (a, d) Spatial distribution of XCH<sub>4</sub> observed by TROPOMI. (b, e) Differences between prior simulations and observations. (c, f) Differences between posterior simulations and observations.

We further conducted an independent evaluation of the posterior estimate by comparing it with 10 *insitu* and flask monitoring sites from the CH<sub>4</sub> ObsPack v6.0 database. Figure 5 shows the time series comparison of the CEP and VEP experiments with the observations. The evaluation statistics for all sites are presented in Table S4. There was a significant overestimation in the CEP experiment

throughout most of the study period. Although most of the sites are located in the eastern regions outside China, influenced by the atmospheric circulation, the westerlies prevail in most parts of China. Thus, the optimized emission information can be effectively reflected in the concentrations observed downwind. The optimized simulation in VEP experiments was more consistent with the observations. Among them, the AMY flask site showed the largest reduction in bias, with the average bias decreasing from 35.3 ppb to 3.3 ppb, a decrease of 90.7%. For WLG, the only ObsPack site available in mainland China, the average bias decreased from 47.4 ppb to 28.3 ppb, a reduction of 40.3%. This indicates that the optimized emissions can significantly improve the CH<sub>4</sub> simulation, whether in areas close to the source or downwind of the source. Overall, the average bias of the 10 sites decreased by 58.6%, from 28.6 ppb to 11.9 ppb, the root mean square error (RMSE) decreased by 28.9%, from 56.2 ppb to 39.9 ppb, and the CORR increased from 0.72 to 0.75. These evaluation results demonstrate that the inversion effectively reduces the uncertainties of prior emission inventory.

**Figure 5** Time series comparison of surface CH<sub>4</sub> concentrations (ppb) from prior (CEP) and posterior (VEP) emission simulations with observations from 5 flask sites and 5 *in-situ* sites within the ObsPack dataset. The black, blue, and red values represent the averaged observations, prior simulations, and posterior simulations, respectively.

We also carried out an independent validation using six high-precision *in-situ* observation sites within Shanxi Province (Figure 1a). Table 2 shows the bias, RMSE, and CORR of the CEP and VEP simulations against these surface observations. Except for the LF site, which shifted to a severe negative bias and exhibited a larger RMSE, the VEP experiment demonstrated varying degrees of

improvement at the other five sites. Notably, the bias reduction at the DT and WTS sites exceeded 93.3%. Especially for the WTS site, a high mountain site with an altitude of over 2208 m, the bias was decreased to 8.3 ppb. On average across all sites, the bias significantly decreased by 96.0%, from 535.1 ppb to 21.4 ppb. For the RMSE, the most significant decreases were observed at the DT and SZ sites, both exceeding 73%. Overall, the RMSE reduction ranged from 13.5% to 79.0%. Additionally, the CORR of the VEP experiment increased to 0.48–0.63. In addition, we evaluated the CH<sub>4</sub> concentration simulation for the afternoon, during which the model generally demonstrates better boundary layer simulation performance, with overall lower bias and higher CORR (Table S5). These results further confirm that the RegGCAS-CH<sub>4</sub> system can effectively capture the characteristics of high-resolution CH<sub>4</sub> emission changes and improve the accuracy of concentration simulations.

**Table 2** Statistics comparing the daily average CH<sub>4</sub> concentrations (ppb) from the simulations with prior (CEP) and posterior (VEP) emissions against six independent surface *in-situ* observation sites in Shanxi Province, respectively. The numbers under the site names represent the number of valid observations.

| Site Name   | Mean<br>Obs. | Mean Sim. |        | BIAS   |        | RMSE   |       | CORR |      |
|-------------|--------------|-----------|--------|--------|--------|--------|-------|------|------|
|             |              | CEP       | VEP    | CEP    | VEP    | CEP    | VEP   | CEP  | VEP  |
| TY<br>(351) | 2595.5       | 2771.0    | 2353.2 | 175.4  | -242.3 | 408.7  | 353.6 | 0.46 | 0.51 |
| DT (286)    | 2122.0       | 2441.1    | 2139.2 | 319.1  | 17.2   | 489.0  | 102.6 | 0.49 | 0.58 |
| LF<br>(277) | 2395.1       | 2397.2    | 2197.9 | 2.0    | -197.3 | 186.8  | 257.3 | 0.45 | 0.35 |
| SZ<br>(359) | 2148.8       | 4560.5    | 2782.4 | 2411.7 | 633.6  | 2778.1 | 745.1 | 0.44 | 0.48 |
| JC<br>(362) | 2446.0       | 2624.5    | 2355.2 | 178.4  | -90.9  | 385.0  | 356.3 | 0.52 | 0.52 |
| WTS (361)   | 2064.0       | 2188.0    | 2072.2 | 124.1  | 8.3    | 162.6  | 59.1  | 0.49 | 0.63 |

<sup>\*</sup> BIAS, mean bias; RMSE, root mean square error; CORR, correlation coefficient

## 4. Discussion

To further explore the characteristics of our posterior emissions and offer valuable guidance for the

refinement of bottom-up inventory in China, we conducted a comparison analysis with the latest 8 bottom-up inventories and 4 top-down emission estimates (Figure 6). Specifically, apart from EDGAR v8, the bottom-up inventories were sourced from the Copernicus Atmosphere Monitoring Service (CAMS-GLOB-ANT v6.2) (Soulie et al., 2024), Community Emissions Data System (CEDS 202407) (Hoesly et al., 2018), Evaluating the Climate and Air Quality Impacts of Short-Lived Pollutants inventory (ECLIPSE v6b) from the GAINS model (Stohl et al., 2015), Global Carbon Project (GCP 2024) (Saunois et al., 2025), Peking University (PKU v2) (Liu et al., 2021), United Nations Framework Convention on Climate Change (UNFCCC, 2020), and the United States Environmental Protection Agency (EPA, 2019). In addition, we also obtained the Global Fuel Exploitation Inventory (GFEI v2) (Scarpelli et al., 2022) to assess the emissions from the coal mining sector in Shanxi Province. The top-down emissions were mainly sourced from the CarbonTracker-CH<sub>4</sub> (Oh et al., 2023) and the research results of Chen et al. (2022), Kou et al. (2025), and Peng et al. (2023). Typically, the top-down estimates showed lower emissions than the bottom-up inventories. Although our posterior estimates were the lowest across all datasets, they closely matched CEDS, PKU, and CarbonTracker-CH<sub>4</sub> and were comparable to the ensemble mean of GCP ground-based inversions (Wang et al., 2025). Overall, our posterior emissions were 22.0% lower than the average of bottom-up inventories and 16.6% lower than the previous top-down estimates. The lower emissions in this study were predominantly driven by the downward revision of coal emissions. Overestimated emission factors and the difficulty in tracking the spatial distribution of coal mines due to mine closures and regional transfers remain significant obstacles to the assessment of coal mine emissions (Gao et al., 2021). Our estimate was lower than the previous top-down studies (53–65 Tg yr<sup>-1</sup>), likely because those previous studies were conducted at much coarser resolutions (0.3°-2.5° versus 27 km) or with much sparser observations (in-situ ground measurements and GOSAT versus TROPOMI). For Shanxi Province, compared with the bottom-up inventories, our results were close to those of GAINS and fell within the uncertainty range of Qin et al. (2024), which set multiple groups of emission factors according to coal types, mining methods, and geological structures. There were substantial disparities among the top-down inversions. Notably, for CarbonTracker-CH4, despite having minimal divergence at the national scale when contrasted with other estimations, the values for Shanxi Province were






























markedly lower than those of other estimates. This could be attributed to the insufficient assimilation

of China's surface observations and systematic biases in the spatial distribution of prior emissions. Overall, the coal-mining emissions in this study were 29.2% lower than the bottom-up inventories and 19.3% lower than the previous top-down inversions.

**Figure 6** Total anthropogenic CH<sub>4</sub> emissions (TgCH<sub>4</sub>·yr<sup>-1</sup>) in China (a) and CH<sub>4</sub> emissions from the coal mining sector in Shanxi Province (b). GCP 2024 BU and GCP 2024 TD represent the bottom-up and top-down ensemble means included in the dataset, respectively. The grey bars in panel b represent the range of CH<sub>4</sub> emissions estimated by Qin et al. (2024) using four different methods. The latest year in the inventory is marked above the bars.

Uncertainties in boundary conditions constitute a significant source of error in regional inversion.

Despite the optimization of concentration fields in the CAMS, significant biases remained evident

(Figure 7a). To address this issue, we implemented additional constraints on the CAMS concentration fields using TROPOMI XCH<sub>4</sub> observations, resulting in a notably improved agreement with the observations. Specifically, the average bias decreased from -3.7 ppb to -1.0 ppb, representing a reduction of 73.7%. A sensitivity experiment (SENS1) was further conducted (Table S6), where the unadjusted CAMS global fields were extracted as boundary conditions to invert anthropogenic CH<sub>4</sub> emissions, aiming to evaluate the impact of boundary condition uncertainties on regional emission inversions. Compared with the base experiment (BASE), the largest discrepancies in monthly variations were observed during the winter months, indicating significant overestimations. The emission differences across different months closely aligned with the concentration differences, suggesting that underestimations in concentrations prompted more substantial emission adjustments for compensation. Regarding the spatial distribution, due to the relative long lifetime of CH<sub>4</sub>, emission changes caused by boundary condition biases were not confined to regional boundaries, such as in Northeast and Northwest China, but were also observable throughout the entire region. In the outer national inversion, the average CH<sub>4</sub> emissions increased by 7.5%, while in the inner Shanxi Province, the increase was 3.9%. This highlights the necessity of expanding the inversion scope to mitigate the influence of boundary condition errors on the inversion of central regions. Furthermore, we compared two sets of emission differences over the inner domain: one between the two national inversions, and the other between the two inner nested regional inversions (BASE and SENS1). We found that the two inner inversions exhibited a 23.1% smaller discrepancy. This indicates that adopting a nested inversion approach is essential, which can further reduce boundary condition errors during inner inversion through outer optimization, thereby enhancing the robustness of the inversion process.























**Figure 7** Differences in XCH<sub>4</sub> column concentrations (ppb) between TROPOMI-adjusted and unadjusted CAMS fields, along with the induced monthly posterior CH<sub>4</sub> emission differences (SENS - BASE) over China (a); differences in two posterior emissions (MgCH<sub>4</sub>·km<sup>-2</sup>·yr<sup>-1</sup>) derived from unadjusted and adjusted boundaries over China (b) and Shanxi Province (c)

To better evaluate the potential impact of prior uncertainties on posterior emission estimates, we conducted additional inversion experiments (SENS2) using the 2022 CAMS-GLOB-ANT v6.2 inventory as prior emissions. Nationwide, the posterior emissions in SENS2 increased by 5.2% compared with those in the BASE experiment. More importantly, the initial difference between the two prior inventories (6.0 Tg) converged to a much smaller difference of 2.3 Tg in the posterior results, indicating good robustness of the assimilation system at the national scale. However, in southern China

(south of 30°N), due to limited observational constraints, the difference between the two prior inventories (5.8 Tg) only decreased to 4.8 Tg in the posterior results. In contrast, in observation-dense regions such as Shanxi Province, even though the difference in prior emissions was only 61.9 Gg, the difference in optimized posterior emissions further converged to 39.0 Gg.

Uncertainty in atmospheric transport models can contribute to model-data mismatch errors. Consistent with previous studies (Chen et al., 2022), this study initially omitted CH<sub>4</sub> chemical reactions to accelerate model integration and inversion efficiency. To quantify the impact of this simplification, we further conducted an additional inversion experiment (SENS3) where CH<sub>4</sub> chemical reactions were incorporated into the CMAQ model. Results showed that the inclusion of chemical reactions led to a 6.6% difference compared to the base experiment. Specifically, the difference was small in winter (only 1.7%), whereas in summer, the OH concentration in the lower troposphere was one order of magnitude higher than that in winter (Lelieveld et al., 2016). This stronger OH-driven CH<sub>4</sub> oxidation resulted in an increase of over 10% in posterior emissions. This indicates that accounting for CH<sub>4</sub> chemical reactions in summer is still necessary for accurate emission inversion. The impact of chemical reactions only increased emission estimates by 1.9% in Shanxi Province.

Different satellite products employ distinct inversion algorithms, which in turn determine the quality and quantity of the data. To assess how satellite product selection influences emission inversion, the TROPOMI/WFMD product was assimilated in SENS4. Compared with the operational TROPOMI product in BASE experiment, the TROPOMI/WFMD product provided a 59.3% increase in the number of observations, particularly notable in winter. In mainland China, posterior emissions derived from SENS4 increased by 4.4%, primarily driven by higher emission estimates in March and April. In Shanxi Province, posterior emissions showed a more modest increase of 2.2%.

Our results may also be subject to several uncertainties associated with the settings of assimilation system parameters. In particular, background and observation errors influence the weight assigned to prior emissions versus observations in determining posterior emissions, while the localization scale dictates the distance over which observational information affects the inversion results. To quantify these impacts, we conducted sensitivity tests by adjusting key parameters: observation errors were set to 0.5% and 0.9% (SENS5-6), background errors to 30% and 50% (SENS7-8), and localization scales to 250 km and 350 km (SENS9-10), respectively. However, our sensitivity analysis revealed that

varying these parameters, whether increasing or decreasing their values, only led to differences of -0.7% to 1.7% in posterior emission estimates across mainland China. This indicates that the CH<sub>4</sub> emission estimates were not significantly affected by adjustments to the system parameters.

Following the methods of Feng et al. (2024) and Nassar et al. (2017), we estimated the overall uncertainty of our results by accounting for the combined effects of the aforementioned factors (e.g., parameter settings, prior inventories). In general, sparsely observed regions, such as western China and Northeast China, showed over-reliance on the prior inventory (SENS2) and exhibited relatively high posterior emission uncertainty (28.0–44.1%). In contrast, densely observed regions including East China and North China showed relatively low uncertainty (7.9–17.4%). Across mainland China, boundary condition errors contributed the most to total uncertainty. Specifically, boundary conditions caused substantial differences in emission estimates for Northeast China. The overall posterior emission uncertainty for mainland China was 8.5%, while that for Shanxi Province was even lower at 7.8%, with this uncertainty primarily driven by uncertainties in the prior inventory.

**Figure 8**. Posterior CH<sub>4</sub> emissions from the base and different sensitivity experiments (Table S6). B denotes the BASE experiment. S1 to S10 denote the SENS1 to SENS10 experiments, respectively. S1 represents the experiment using the unadjusted CAMS global concentration field as the boundary; S2 denotes the experiment adopting CAMS-GLOB-ANT v6.2 inventory as the prior emission inventory; S3 denotes the inversion experiment accounting for CH<sub>4</sub> chemical reactions; S4 denotes the inversion experiment assimilating the TROPOMI/WFMD product; S5 and S6 denote the experiments with observation errors set to 0.5% and 0.9%, respectively; S7 and S8 denote the experiments with background errors set to 30% and 50%, respectively; S9 and S10 denote the experiments with the localization scale adjusted to 250 km and 350 km, respectively. The numbers on the figure represent the uncertainty values of different regions.

Our inversion results are generally lower than previous emission estimates. By comparing the emission differences over the inner region at different resolutions across the BASE and all SENS experiments, we found that emissions inverted at a 9 km resolution were typically 5.4-10.6% lower than those at a 27 km resolution. This indicates that higher-resolution inversion consistently yields lower emission estimates, a discrepancy likely driven by the fact that higher-resolution simulations excel at capturing

localized emission hotspots that lead to elevated concentration values. Therefore, compared with other coarse-resolution inversions, this study tends to adjust emissions downward to better align with observations. To further understand the extent to which different sensitivity factors affect our relatively low posterior emission results, we compared the emission differences between SENS experiments and the BASE experiment under the inner domain coverage at the same 27 or 9 km resolution (Figure S7). In every sensitivity experiment with a positive difference (SENS – BASE), the magnitude varies between 0.3-7.2%, which is always smaller than the emission reduction caused by high-resolution inversion under the corresponding SENS experiment. This confirms that higher resolution remains the dominant driver of our lower inversion results relative to previous studies, while the aforementioned factors contribute to secondary, manageable uncertainties. Additionally, China's 2022 COVID-19 restrictions, the most stringent since 2020, may be another factor driving lower emissions due to production shutdowns and home quarantine. Peng et al. (2022) combined bottom-up and top-down approaches to quantify CH<sub>4</sub> source changes, revealing a 1.2 Tg·CH<sub>4</sub> yr<sup>-1</sup> reduction in anthropogenic CH<sub>4</sub> emissions in 2020 compared to 2019. Taking oil and gas sector as an example, an IEA report stated that global CH<sub>4</sub> emissions from oil and gas operations decreased by approximately 10% yearon-year in 2020 (IEA, 2021), primarily attributed to reduced oil and gas production during the pandemic (Thorpe et al., 2023).

## 5 Summary and conclusions



















- In this study, we developed a Regional Methane Assimilation System (RegGCAS-CH<sub>4</sub>) based on the
- WRF-CMAQ model and EnKF algorithm to perform high-resolution nested inversions of China's CH<sub>4</sub>
- emissions using TROPOMI XCH<sub>4</sub> retrievals. Taking the EDGAR v8 as prior emissions, we inferred
- the daily anthropogenic CH<sub>4</sub> emissions over China in 2022.
- Using TROPOMI XCH<sub>4</sub> as constraints, we revealed significant overestimations in the prior emissions.
- Nationally, the posterior anthropogenic CH<sub>4</sub> emissions in 2022 were estimated to be  $45.1 \pm 3.8$
- TgCH<sub>4</sub>·yr<sup>-1</sup>, 36.5% lower than the EDGAR inventory, mainly driven by the decreases in the coal and
- waste sectors. North China experienced the most substantial reduction, with a 56.9% decrease, mainly
- due to downward revisions in the coal and enteric fermentation sectors. In Shanxi Province, influenced
- by dominant sources of coal mining, the posterior emissions decreased by 46.3% compared to the
- EDGAR inventory, with value of  $4.8 \pm 0.4 \text{ TgCH}_4 \cdot \text{yr}^{-1}$ . The monthly variation of posterior emissions

showed a pattern of being low in winter and high in summer. Except for the rice cultivation and waste sectors, which showed some increases in certain months related to seasonal growth stages and temperature-driven methanogen activity, respectively, other sectors exhibited varying degrees of reduction. Observation-constrained emissions significantly improved the CH<sub>4</sub> simulation performance. Specifically, for the entire region and Shanxi Province, the biases were reduced by 46.0% and 68.7% for XCH<sub>4</sub> columns, and by 58.6% and 96.0% for surface CH<sub>4</sub>, respectively. This highlights the effectiveness of RegGCAS-CH<sub>4</sub> in reducing uncertainty in CH<sub>4</sub> emissions. Sensitivity inversions highlight the importance of high-resolution satellite-based nested inversions in accurately estimating CH<sub>4</sub> emissions, especially in reducing the impact of boundary condition errors on emission inversions. Top-down inversions usually result in lower emissions compared to those constructed by bottom-up methods. When comparing our posterior emissions with other inventories, our estimates were the lowest among both bottom-up and top-down estimations. On average, they were 22.0% lower than bottom-up inventories and 16.6% lower than previous top-down estimates. The high-resolution inversion employed in this study is capable of more accurately simulating local high concentration values, and thus, the reduction in emissions is greater than that in previous studies. The overestimation of China's CH<sub>4</sub> emissions may have led to an overestimation of the country's climate change mitigation burden. The more accurate emission estimates presented here not only enhance our understanding of the CH<sub>4</sub> budget but also contribute to more effective global climate change mitigation efforts.

773774


















## Data availability

- The TROPOMI XCH<sub>4</sub> data is publicly available for download at the Copernicus Data Space Ecosystem
- (https://browser.dataspace.copernicus.eu/, last access: August 5, 2024). The observations used for
- evaluation and the posterior emissions and simulations produced in this study can be accessed at
- https://doi.org/10.5281/zenodo.15602944 (Feng, 2025).

## **Author contribution**

- SF, FJ and YZ conceived and designed the research. SF developed the RegGCAS-CH<sub>4</sub> system,
- analyzed data, and prepared the paper. SW evaluated the CH<sub>4</sub> simulation in Shanxi Province. HC, HZ,
- SB, HW, and WJ reviewed and commented on the paper.

## **Competing interests**

- At least one of the (co-)authors is a member of the editorial board of Atmospheric Chemistry and
- Physics.

785

## Acknowledgements

- This work is supported by the National Key R&D Program of China (Grant No: 2022YFE0209100),
- the National Natural Science Foundation of China (Grant Nos. 42305116 and U24A20590), the
- Natural Science Foundation of Jiangsu Province of China (Grant No. BK20230801), the 'GeoX'
- Interdisciplinary Project of Frontiers Science Center for Critical Earth Material Cycling (Grant No.
- 20250104), and the Fundamental Research Program of Shanxi Province (Grant No.
- 202403011232002). We thank Professor Euan G. Nisbet for his valuable feedback and constructive
- comments that enhanced the clarity and quality of this paper. We also gratefully acknowledge the High-
- Performance Computing Center (HPCC) of Nanjing University for supporting the numerical
- calculations in this paper on its blade cluster system.

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
