# Peer review of "High-resolution regional inversion reveals overestimation of anthropogenic 1 methane emissions in China 2 Shuzhuang Feng1,2, Fei Jiang1,2,5\*, Yongguang Zhang1,2, Huilin Chen3, Honglin Zhuang1, Shumin 3 Wang3,4, Shengxi Bai1, Hengmao Wang1,2, Weimin Ju1,2 4 5 1 Jiangsu Provincial Key Laboratory of Geographic Information Science and Technology, 6 International Institute for Earth System Science, Na"

_EGUsphere, 2025_

## Author Comment (AC1)

**Responses to the comments of Reviewer #1:**

We would like to thank the anonymous referee for his/her comprehensive review and valuable suggestions. These suggestions help us to present our results more clearly. In response, we have made changes according to the referee's suggestions and replied to all comments point by point. All the page and line number for corrections are referred to the revised manuscript, while the page and line number from original reviews are kept intact.

**Main comments:**

**1.** The study discussed methane emissions from different sectors, but how the sector partitioning is done is not described. I'd suggest authors to provide further methodological details.

**Response:** We appreciate the reviewer for the insightful comments. Actually, the observed total atmospheric $CH_4$ concentration integrates emission signals from all sectors, making it difficult to distinguish emission information from different source sectors overlapping in a pixel grid (Saunois et al., 2025). Therefore, the emissions derived from inversion are generally the total emissions at the pixel scale. Following Kou et al. (2025), Zhang et al. (2022), and Miller et al. (2019), we partitioned the optimized total emissions based on the prior proportional information of different sectors within the same model grid. However, it is true that errors in the sectoral proportions of the prior inventory introduce uncertainties into the posterior statistics.

We have added following descriptions and discussions in the revised manuscript. See Lines 377-381, Pages 16-17.

"Assimilating total $CH_4$ observations alone cannot disentangle emissions from different source sectors overlapping in individual grid cells (Saunois et al., 2025). Consequently, we partitioned the inversion results into respective emission sectors based on the monthly prior proportions at the model grid points (Kou et al., 2025; Zhang et al., 2022), though this approach does introduce a certain degree of uncertainty in sectoral

attribution. The sectoral patterns offer insights into the underlying factors influencing China's emission changes. We concentrated on interpreting the emissions from the coal, gas, rice cultivation… …"

**2.** The authors have applied TROPOMI XCH4 L2 data. An earlier version of the TROPOMI data have shown substantial regional biases over East China, which may cause errors in the inversion. It would be good if the authors can have some discussion or conduct evaluation on this issue, for instance, using TCCON sites in China.

**Response:** Thanks for your comments. Indeed, the TROPOMI $XCH_4$ L2 data product used in this study is Version 02.04.00, and we also identified a considerable number of unrealistically low values in the raw data, particularly during summer (Figure R1). To evaluate the data quality, we compared the raw TROPOMI data with observations from two domestic TCCON stations in China (i.e., Hefei Station and Xianghe Station). The results showed that the raw TROPOMI data underestimated the $XCH_4$ concentration by 13.2 ppb and 7.8 ppb at these two stations, respectively. This magnitude of negative bias is comparable to the global evaluation results based on 12 TCCON stations, which reported a bias range of -8.5 to -15.5 ppb (PRF-CH4, https://sentiwiki.copernicus.eu/web/s5p-products#S5PProducts-L2S5P-Products-L2). Such negative biases, if unaddressed, would inevitably lead to the underestimation of inverted emissions. However, to avoid the impact of such negative biases on the inversion results, we not only excluded pixels with a qa_value below 0.5 but also selected an alternative $XCH_4$ product (derived from the WFMD algorithm) to conduct cross-validation. Only the data that met both of the aforementioned criteria were used in the final assimilation. Figure R2 displays the time series of the data after final quality control. It can be observed that this data aligns well with TCCON observations, with relatively small overall differences.

We have added following discussions in the revised manuscript. See lines 294-300, Pages 12-13.

"However, we still found many unrealistic low values, especially in summer. These negative biases can inevitably lead to the underestimation of inverted emissions. To further minimize the impact of outliers, … …Only those pixels that were concurrently available in the TROPOMI/WFMD product and met the quality flag requirements were assimilated. Subsequently, the final quality-controlled TROPOMI data demonstrated good consistency with observations from two TCCON stations, namely the Hefei and Xianghe stations (Figure S3)."

[Figure]

**Figure R1** Comparison of time series between operational TROPOMI XCH₄ product filtered by qa_value > 0.5 and TCCON observations at Hefei and Xianghe stations. For the evaluation, only TROPOMI pixels that are located within a 0.1° radius of the respective TCCON station and have a time difference of less than 1 hour relative to TCCON observational records (two spatiotemporal matching criteria) were selected. Specifically, the number of valid matching pairs was 62 for the Hefei station and 163 for the Xianghe station.

[Figure]

**Figure R2** Same as Figure R1, but for the evaluation of TROPOMI data after final quality control. The number of valid matching pairs was 57 for the Hefei station and 155 for the Xianghe station. (Figure S3 in the revised supplementary information)

**3.** I do appreciate that the authors have performed evaluation for meteorology parameters against independent data, which most of existing studies have not done. This is crucial for characterizing model transport errors and understanding the difference between inversion systems. However, the discussion is overly simple. I'd suggest the authors to expand the results on meteorology evaluation (especially wind). In particular, the evaluation over the D02 domain provides crucial information because of the complex terrain in Shanxi.

**Response:** We fully agree that $CH_4$ emission estimates are highly sensitive to biases in meteorological simulations. This is because meteorological processes exert a significant influence on atmospheric transport, which in turn shapes the source-receptor relationships and determines the flow-dependent background error covariance. As shown in Figure R3, we expanded the meteorological evaluation with a specific focus on wind conditions and incorporated an assessment of the meteorological field simulations over Shanxi Province. Overall, across the China domain, the WRF model simulations exhibited biases of -0.4°C for T2, -4.9% for RH2, and 0.5 m/s for WS10. For Shanxi Province, which features complex terrain, the biases were 1.5°C for T2, -12.5% for RH2, and 0.4 m/s for WS10. Notably, the overestimated wind speed in the simulations accelerated the transportation of simulated $CH_4$ concentrations, which to some extent contributed to the overestimation of inverted $CH_4$ emissions.

Relevant discussions have been added to the revised manuscript. See Lines 523-537, Pages 22-23.

"$CH_4$ emission estimates are highly sensitive to biases in meteorological simulations, as meteorological processes significantly influence atmospheric transport, which in turn shapes the source-receptor relationships and determines the flow-dependent background error covariance. Overall, the model demonstrated satisfactory performance in reproducing domain-wide meteorological conditions across China (Figure S6), with minimal biases of -0.4°C for T2 and -4.9% for RH2, alongside high correlation coefficients (CORR) of 1.0 and 0.96, respectively. The model's performance over Shanxi Province was slightly less optimal, likely due to the region's complex terrain. The biases for T2 and RH2 were 1.5°C and -12.5%, respectively, while the CORR remained high at 0.99 and 0.90. Additionally, the WRF model effectively captured the temporal variations in wind direction both across China and Shanxi Province. However, WS10 was generally slightly overestimated, with biases of 0.5 m/s and 0.4 m/s and CORR of 0.88 and 0.81 over China mainland and Shanxi province, respectively. Such overestimation of wind speed in WRF simulations has also been widely reported in other studies (e.g., Hu et al., 2016). An overestimated wind speed

causes the model to simulate faster and more extensive diffusion of CH₄ concentrations than occurs in reality. To compensate for the simulated reduction in CH₄ concentrations due to this excessive diffusion, the inversion system potentially increases the estimated emissions."

[Figure]

**Figure R3** Time series of observed and simulated wind direction at 10 m (WD10), wind speeds at 10 m (WS10, m/s), temperature at 2 m (T2, °C), and relative humidity at 2 m (RH2, %) across (a-d) China and (e-h) Shanxi Province. China and Shanxi Province include 400 and 26 stations, respectively. (Figure S6 in the revised supplementary information)

**4.** The authors used the optimized emissions from the D01 inversion as prior emissions for the D02 inversion. This implies that the observations over D02 are used twice in the optimization of emissions. From the Bayesian standpoint, this is

problematic as it leads to over-confidence in observations.

**Response:** Thanks for this comment. It is important to clarify that in the first inversion window, the prior emissions for both the D01 and D02 domains were derived from the EDGAR inventory. For subsequent inversion windows, the prior emissions ($X_b$) for D01 and D02 were each obtained from their respective optimized emissions ($X_a$) in the previous window (Figure R4). Notably, the observational data were used sequentially to optimize the prior emissions of D01 and D02. Specifically, the observations were not reused for the optimization of the same emissions. Instead, D01 only provided a more optimized boundary condition for the emission simulation of D02, rather than serving as the prior emission input for D02.

To enhance clarity on this process, we have added the following description in the revised manuscript. See Lines 152-161, Pages 6-7.

"… …For the same domain, the RegGCAS-CH$_4$ performed a "two-step" inversion scheme in each data assimilation (DA) window. First, the prior emissions were optimized using the available atmospheric observations. Then, the optimized emissions were input back into the CTM to generate the initial fields for the next assimilation window. Simultaneously, the optimized emissions were transferred to the next window to serve as prior emissions (Figure S1). It is noted that the system optimizes the prior emissions for the D01 and D02 domains separately. Specifically, D01 only provides an optimized boundary field for D02, rather than the prior emission source for D02. Thus, the uncertainties in boundary conditions for D02 emission estimates were reduced. … …"

[Figure]

**Figure R4** RegGCAS-CH4 assimilation process (Figure S1 in the revised supplementary information)

**5.** The paper in general lacks uncertainty characterization for emission flux estimates. For regions with limited observation coverage (e.g., Southern China), it is unclear to what degree the posterior estimates depend on prior estimates.

**Response**: Thank you for this insightful comment. To quantify the overall posterior uncertainty of our emission estimates, we evaluated the combined impacts of multiple influential factors, including the representation of $CH_4$ chemical processes, boundary conditions, selection of prior inventories, satellite products, and assimilation system parameter settings (Figure R5). Meanwhile, we paid special attention to southern China (a region with limited observation coverage) to assess the degree to which posterior

emissions rely on prior estimates.

Nationwide, adopting the CAMS-GLOB-ANT v6.2 inventory (instead of the base EDGAR inventory) led to a 5.2% increase in posterior emissions. More importantly, the initial difference between the two prior inventories (6.0 Tg) converged to a much smaller difference of 2.3 Tg in the posterior results, indicating good robustness of the assimilation system at the national scale. However, in southern China (south of 30°N), limited observational constraints weakened this convergence. The difference between the two prior inventories (5.8 Tg) only decreased to 4.8 Tg in the posterior emissions, clearly reflecting a higher dependence of posterior estimates on prior information in this region. Using the default CAMS global concentration field (relative to adjusted fields) resulted in a 7.5% increase in posterior emissions; incorporating $CH_4$ chemical reactions (vs. omitting them) caused a 6.6% increase; assimilating the TROPOMI/WFMD product (vs. the TROPOMI/SRON product) led to a 4.4% increase. In contrast, variations in assimilation system parameters (e.g., observation error, background error, and localization scale) had minimal impacts, restricting changes in posterior emissions to a narrow range of -0.8% to 1.7%. Based on these analyses, we quantified the overall posterior emission uncertainty as 8.5% for mainland China and 7.8% for Shanxi Province.

We have added the above discussion on uncertainty characterization in the revised manuscript.

See Lines 676-736, Pages 31-34.

[revised manuscript text omitted]

**Minor suggestions:**

1. L80: key source-> "point source scale" or "local scale"?

**Response**: We have changed "key source scales" to "local scales". See Line 80, Page 4.

2. L95: unclear -> uncertain

**Response**: We have changed "unclear" to "uncertain". See Line 95, Page 4.

3. L103: To my knowledge, IMI is not an operational inversion system, but more like open-source software. So it may be improper to characterize it as a US system. Similar issues may exist for other listed systems.

**Response**: Thank you for pointing out this important issue. we have revised the description to accurately reflect its characteristics. See Lines 99-105, Pages 4-5.

"Currently, global-scale $CH_4$ assimilation systems are widely applied, such as CarbonTracker-$CH_4$ in the United States (Bruhwiler et al., 2014), CAMS in Europe (Agustí-Panareda et al., 2023), NTFVAR in Japan (Wang et al., 2019), and GONGGA-$CH_4$ in China (Zhao et al., 2024)… …There are relatively few existing regional $CH_4$ assimilation systems, such as the ICON-ART-CTDAS (Steiner et al., 2024) and CarbonTracker Europe-$CH_4$ (Tsuruta et al., 2017) in Europe. Additionally, several open-source frameworks offer inversion tools adaptable to different scales, such as LMDz-SACS-CIF in France (Thanwerdas et al., 2022) and the IMI in the United States (Varon et al., 2022). Nevertheless, most existing regional inversions still rely on global atmospheric transport models with relatively coarse resolutions and… …"

4. L188-189: Any quantitative estimates how much error it will incur for D01 and for D02 respectively, by deactivating the chemical oxidation?

**Response**: Thank you for this valuable comment. To address your question regarding the quantitative error in CH$_4$ emission estimates caused by deactivating CH$_4$ chemical oxidation, we conducted a dedicated sensitivity experiment (SENS3) where full CH$_4$ chemical reactions were incorporated into the CMAQ model. For the mainland China, the omission of CH$_4$ chemical reactions results in an overall underestimation of posterior emissions by approximately 6.6% compared to the SENS3 experiment (with reactions activated). For the Shanxi Province, the bias induced by deactivating chemical oxidation is more modest, with an average underestimation of only 1.9% across all seasons. We have supplemented error estimates in the revised manuscript. Please refer to our response to Main Comment 5.

5. L230: How do you specify the R matrix? Also explain specifically that R is an error covariance matrix for what.

**Response**: We sincerely appreciate your meticulous review. The matrix $\boldsymbol{R}$ is an observation error covariance matrix. It is specified as a diagonal matrix, which assumes that observation errors from different stations at different times are mutually independent (i.e., no covariance between distinct observations). The diagonal elements correspond to the observation errors of the satellite data, set here to be 0.7% (~ 13.3 ppb in mainland China) of the column concentration values. This specification is based on the product's quarterly validation report, which indicates that for the bias-corrected TROPOMI product, the 1σ spread of the relative difference between TROPOMI retrievals and TCCON observations is on the order of 0.7% (Lambert et al., 2025).

We have added a description of the $\boldsymbol{R}$ matrix. See Lines 232-233, Page 10.

"where $\boldsymbol{R}$ is an observation error covariance matrix, which is specified as a diagonal matrix with the assumption that observation errors from different pixels are mutually independent (Feng et al., 2020). $\boldsymbol{K}$ is the Kalman gain matrix … …"

6.   L232: Ep: Power plant sources? Seems something copied from a CO2 study.

**Response**: Thank you for your comment. Indeed, the prior inventory (EDGAR) used in this study includes the "Power Industry" sector (ENE). Given that power plants are typically elevated point sources, they are usually not located in the same model grid as ground-based area sources. This spatial distinction allows for effective separation between these two types of sources. Therefore, even though power plant sources account for a small proportion (0.6%) of total emissions, we treated area sources and power plant sources as separate state vectors for optimization in the inversion process.

We have added additional explanations. See Lines 239-241, Page 11.

"… …industry, transport sources, etc. Given that power plants are typically elevated point sources, this spatial distinction allows for effective separation from ground-based area sources. Therefore, even though power plant sources account for a small proportion (0.6%) of total emissions, we treated them as separate state vectors for optimization. The updated emissions are then… …"

7.   L234: No need to capitalize O in oil

**Response**: The capitalization of "O" in "oil" has been corrected. Thanks. See Line 238, Page 10.

L251: Would 1 day be too short for adequate observation constraint, if you assume that prior errors are independent from one day to the next (L272-273)?

**Response**: Thank you for this comment. In fact, the prior errors across different inversion windows are not independent, as the prior emissions for each day are derived from the optimized emissions of the previous day. The 40% uncertainty setting is intended to cover the error statistical characteristics of emission variations from one day to the next. This temporal continuity ensures that prior errors do not become completely decoupled between consecutive days.

Theoretically, a longer inversion window would allow $CH_4$ to undergo more extensive atmospheric transport, enabling more observations to capture the signal of emission changes in a given grid cell. However, as the distance between an observation site and an emission source increases, the emission signal detected by the observation weakens significantly, while noise interference intensifies. Particularly, constrained by the EnKF method with a limited ensemble size, this weakened emission signal tends to be masked by unphysical signals (unrealistic long-distance spurious correlations). Consequently, a longer inversion window does not necessarily yield better performance than a shorter one (Jiang et al., 2021). On the other hand, the TROPOMI satellite provides relatively dense observational data. Even with a short assimilation window (e.g., 1 day), the abundant observations can still effectively capture meaningful emission signals from surrounding grid cells, which is sufficient to optimize regional-scale $CH_4$ emissions. In contrast, for sparse observational data, a longer assimilation window is typically required to capture emission signals from distant sources.

We have calculated the average number of surrounding observations (all quality-controlled pixels falling into the same grid are averaged into a single observation) that each grid is constrained by per day (Figure R6). Overall, most grids in northern China can be constrained by over 40 observations, while most grids in southern China can be constrained by approximately 10 observations. Additionally, during the assimilation process, we filtered out observations with a correlation coefficient < 0.27 (low significance, with p > 0.05) between the emission ensemble and the concentration ensemble at the observation locations. As a result, in the southwest region, some grids are not constrained by observations, accounting for approximately 4.8% of the total grids nationwide; however, the emissions from these unconstrained grids constitute less than 0.0004% of the national total emissions. Therefore, in most parts of the country, a 1-day assimilation window can provide adequate observation constraint.

We have added additional explanations. See Lines 305-309, Page 13.

"For regions with limited observation coverage (e.g., southern China), posterior emission estimates may rely heavily on prior information (see Discussion). On one hand,

the system optimizes emissions in grids surrounding observations through the source-receptor relationship of atmospheric transport, allowing it to impose extensive constraints on emissions (Figure S4); on the other hand, … …"

[Figure]

**Figure R6** Average number of observations constraining each grid per day. (Figure S4 in the Supplementary Information)

8. Table 1: What do the last two columns (building, mature) stand for?

**Response**: Thank you for pointing out this ambiguity. "Mature" in the table is a spelling error and should be corrected to "Manure", which corresponds to the "Manure management" sector, a key source of $CH_4$ emissions from agricultural activities. The "building" represents emissions from small-scale non-industrial stationary combustion, including fuel combustion for heating, cooking, or other energy uses in residential, commercial, or small non-industrial buildings.

We have corrected the typo ("Mature" → "Manure") and added brief annotations for

both columns in the revised Table 1. See Lines 441-443, Page 19

"* Waste includes wastewater treatment, solid waste landfills, and solid waste incineration; Building represents emissions from small-scale non-industrial stationary combustion; Manure refers to emissions from the manure management sector."

9. Table 1 and related discussion (e.g., L360): EDGAR v8.0 is used as prior information. Recent studies have shown that EDGAR has large errors in the spatial and seasonal distribution in rice emissions (Chen et al., 2025; Liang et al., 2024). I'd suggest the authors to briefly discuss the impact on emission quantification and sector attribution in Northeast and East China.

[revised manuscript text omitted]

---

## Author Comment (AC2)

**Responses to the comments of Reviewer #2:**

We would like to thank the anonymous referee for his/her comprehensive review and valuable suggestions. These suggestions help us to present our results more clearly. In response, we have made changes according to the referee's suggestions and replied to all comments point by point. All the page and line number for corrections are referred to the revised manuscript, while the page and line number from original reviews are kept intact.

1. Lines 33–35 and 548–550: Regarding the conclusion that this study's top-down emission inversion is lower compared to others, the authors mainly attribute this to the higher resolution used here, which captures finer emission details—a point supported by comparing results from the inner and outer nested domains. While this explanation is reasonable, it may not be entirely sufficient. Other factors like differences in methods, models, and observations—including how chemical processes, soil sinks, and boundary conditions are represented—could also contribute significantly. It would be good if the authors could briefly discuss these aspects in the Discussion.

**Response**: Thank you for this insightful comment. We agree that factors beyond spatial resolution, including representations of chemical processes, soil sinks, and boundary conditions, could influence top-down emission inversion results. First, regarding soil sinks, the flux of $CH_4$ uptake by soils in China is relatively small. According to the Global Methane Budget report, the latest bottom-up soil sink results for China in 2020 are 1.9-2.3 $Tg \cdot yr^{-1}$, while the top-down inversion results are 1.2-2.7 $Tg \cdot yr^{-1}$ (Saunois et al., 2025). Thus, uncertainties associated with soil sinks have a negligible influence on China's overall $CH_4$ emission estimates.

We also evaluated the impacts of other key factors on inversion results through sensitivity experiments (Figure R1). Using the default CAMS global concentration field (relative to adjusted fields) leads to a 7.5% increase in posterior emissions; adopting the CAMS-GLOB-ANT v6.2 inventory (instead of the base EDGAR inventory) as a

prior results in a 5.2% increase in posterior emissions; incorporating $CH_4$ chemical reactions (vs. omitting them) causes a 6.6% increase in posterior emissions; assimilating the TROPOMI/WFMD product (vs. TROPOMI/SRON) leads to a 4.4% increase in posterior emissions; and variations in assimilation system parameters (e.g., observation error, background error and local scale) have minimal impacts, limiting posterior emission changes to a range of -0.8% to 1.7%.

To further understand the extent to which different sensitivity factors affect our relatively low posterior emission results, we first compared the inversion results of the same experiment under the D02 domain coverage at different resolutions. Overall, we found that in the BASE experiment and all SENS experiments, emissions inverted at a 9 km resolution were typically 5.4–10.6% lower than those inverted at a 27 km resolution. This indicates that higher-resolution inversion consistently yields lower emission estimates. Second, we compared the emission differences between SENS experiments and the BASE experiment under the D02 domain coverage at the same 27 or 9 km resolution. It is important to note that only positive differences (i.e., SENS emissions > BASE emissions) can indicate that this factor, unaccounted for in the present study (i.e., the BASE scenario), may explain the low emission results observed in this study. For example, Figure R2 shows that, for grids within the D02 domain coverage, the emissions inverted in the SENS1 experiment increased by 5.1% and 4.3% compared with those in the BASE experiment at 27 km and 9 km resolutions, respectively. However, under the same D02 domain coverage (i.e., for the same set of grids), SENS1 showed that emissions inverted at 9 km resolution are 8.4% lower than those inverted at 27 km resolution. Overall, in every sensitivity experiment with a positive difference (SENS – BASE: 0.3–7.2%), the magnitude of this positive difference was smaller than the emission reduction caused by high-resolution inversion (9 km – 27 km: 5.4–10.6%) under the corresponding SENS experiment. This confirms that higher resolution remains the dominant driver of our lower inversion results relative to previous studies, while the aforementioned factors (chemical processes, boundary conditions, etc.) contribute secondary, manageable uncertainties. To address

this concern comprehensively, we have added the above discussion in the revised manuscript.

[revised manuscript text omitted]

**Figure R2** Comparison of emission differences under the D02 domain coverage: one between 9 km and 27 km resolutions within the same BASE or SENS experiment (red), and the other between corresponding SENS and BASE experiments at either 27 km (dark blue) or 9 km (light blue) resolution. Note that only positive differences (i.e., SENS > BASE emissions) can indicate unconsidered factors that might lead to the low emission results in our study.   (Figure S7 in the Supplementary Information)

2.    Figure 1b: It shows that TROPOMI data has a high missing rate across the country, especially in the south, with some areas having coverage for only 10% of the dates. For regions with long periods of no observation, how are the posterior emissions represented? Also, what is the impact of this representation on daily, monthly and yearly CH4 emission estimates? It would be helpful if the authors could discuss this as well.

**Response:** Thank you for this comment. First, regarding the representation of posterior emissions in observation-sparse regions, our assimilation system addresses data gaps

through two key mechanisms. On one hand, the system not only optimizes grids with direct observations but also uses the atmospheric transport model to capture the source-receptor relationship between emissions from surrounding grids and $CH_4$ concentrations at observation sites, thereby enabling the optimization of emissions in surrounding grids within a 300 km localization scale. On the other hand, we adopt an iterative approach where emissions optimized in the current assimilation window serve as prior emissions for the next window, facilitating rolling assimilation to sustain the influence of observational information over time.

Therefore, at the daily scale, grids without observational constraints directly adopt emissions from the previous window. This approach may slightly underestimate short-term emission fluctuations but maintains temporal continuity in emission trends. At the monthly scale, grids with no continuous observational constraints throughout the month directly use EDGAR data. Such grids account for 7.9% of all grids and contribute 0.3% to total posterior emissions. While this may lead to insufficient observational constraints on posterior emissions, particularly in southern regions during summer, it effectively avoids seasonal distortions in posterior estimates caused by variations in emissions. At the annual scale, 4.8% of grids remain unadjusted, and the unadjusted emissions in these grids are mainly distributed in uninhabited areas of Southwest China, resulting in a negligible overall impact (0.00037%) on annual $CH_4$ emission estimates.

To further verify the robustness of our emission optimization under limited observations, we conducted additional sensitivity experiments (e.g., SENS2, detailed in Response to Comment 1) where we compared the impact of different prior inventories on posterior emission estimates. We have added the following discussion in the revised manuscript. See Lines 305-319, Page 13.

"For regions with limited observation coverage (e.g., southern China), posterior emission estimates may rely heavily on prior information (see Discussion). On one hand, the system optimizes emissions in grids surrounding observations through the source-receptor relationship of atmospheric transport, allowing it to impose extensive constraints on emissions (Figure S4); on the other hand, it adopts an iterative approach

where emissions optimized in the current window serve as prior emissions for the next window, facilitating rolling assimilation and thereby sustaining the influence of observational information on emission estimates. However, intermittent observations may cause posterior emissions to underestimate short-term emission fluctuations. At the monthly scale, grids without continuous observational constraints throughout the month directly use EDGAR data. Such grids account for 7.9% of all grids and contribute 0.3% to total posterior emissions. Although this may lead to insufficient observational constraints on posterior emissions, particularly in southern regions during summer, it effectively avoids seasonal distortions in posterior estimates caused by variations in emissions. At the annual scale, 4.8% of grids remain unadjusted. These unadjusted emissions are mainly distributed in uninhabited areas of Southwest China, resulting in a negligible overall impact on annual $CH_4$ emission estimates."

3.   Figure 3 on page 20 shows the differences in prior and posterior emissions for different sectors. How are the sectors distinguished in the posterior emissions? It is indeed challenging to differentiate sectoral emissions in top-down emission inversion. Typically, sectoral emissions in the posterior are calculated based on the proportional grid emissions from the prior inventory. Does this study use the same method?

**Response:** Thank you for this comment. Yes, that's correct. Following Kou et al. (2025), Zhang et al. (2022), and Miller et al. (2019), we partitioned the optimized total emissions based on the prior proportional information of different sectors within the same model grid. This method is adopted because the observed total atmospheric $CH_4$ concentration integrates emission signals from all sectors, making it difficult to distinguish emission information from different source sectors overlapping in a pixel grid (Saunois et al., 2025). As a result, the emissions derived directly from our inversion represent the total $CH_4$ flux at the pixel scale. We acknowledge that uncertainties may be introduced into posterior sectoral statistics by potential errors in the sectoral proportionality of the prior inventory, and this limitation is noted in the revised manuscript.

"Assimilating total $CH_4$ observations alone cannot disentangle emissions from different source sectors overlapping in individual grid cells (Saunois et al., 2025). Consequently, we partitioned the inversion results into respective emission sectors based on the monthly prior proportions at the model grid points (Kou et al., 2025; Zhang et al., 2022), though this approach does introduce a certain degree of uncertainty in sectoral attribution. The sectoral patterns offer insights into the underlying factors influencing China's emission changes. We concentrated on interpreting the emissions from the coal, gas, rice cultivation… …"

4. The title of Figure 4 needs to specify the time period covered by the data. Is it the average for the entire year of 2022?

**Response:** Yes. Figure 4 presents the evaluation of $XCH_4$ averaged over the entire year of 2022. We have supplemented the time information in the figure caption to clarify this detail.

"**Figure 4** Comparison of simulated $XCH_4$ (ppb) from prior and posterior emissions with TROPOMI observations over (a-c) China and (d-f) Shanxi Province for the 2022 annual average."

5. On line 509, P24, 'Except for LF site...', this description is incorrect. Not only is the LF site underestimated, but the TY site is also significantly underestimated, as well as the JC site. It is recommended to correct the description."

**Response:** Thank you for pointing out this ambiguity in our description. The original intent was to highlight that the LF site showed a worsened bias, shifting to a severe negative bias and larger RMSE in VEP experiment (posterior simulations). Although the TY and JC sites still showed an underestimation in the VEP experiment, their overall

performance has improved in VEP experiment.

We have revised the sentence. See Lines 588-589, Page 26.

"Except for the LF site, which shifted to a severe negative bias and exhibited a larger RMSE, the VEP experiment demonstrated varying degrees of improvement at the other five sites."